# Enhancing Healthcare Outcomes and Modulating Apoptosis- and Antioxidant-Related Genes through the Nano-Phytosomal Delivery of Phenolics Extracted from *Allium ampeloprasum*

**DOI:** 10.3390/genes14081547

**Published:** 2023-07-28

**Authors:** Ali Shoeibi, Ehsan Karimi, Mohsen Zareian, Ehsan Oskoueian

**Affiliations:** 1Department of Biology, Mashhad Branch, Islamic Azad University, Mashhad, Iran; 2Department of Life Sciences, Chalmers University of Technology, Göteborg, Sweden; 3Industrial and Mineral Research Center, Arka Industrial Cluster, Mashhad, Iran

**Keywords:** nano-drug delivery systems, natural bioactive compounds, anticancer properties, nano-phytosomes, phenolic rich fraction, breast cancer treatment

## Abstract

The application of nano drug delivery systems, particularly those utilizing natural bioactive compounds with anticancer properties, has gained significant attention. In this study, a novel nano-phytosome-loaded phenolic rich fraction (PRF) derived from *Allium ampeloprasum* L. was developed. The antitumor activity of the formulation was evaluated in BALB/c mice with TUBO colon carcinoma. The PRF-loaded nano-phytosome (PRF-NPs) exhibited a sphere-shaped structure (226 nm) and contained a diverse range of phenolic compounds. Animal trials conducted on TUBO tumor-bearing mice demonstrated that treatment with PRF-NPs at a dosage of 50 mg TPC/Kg/BW resulted in significant improvements in body weight and food intake, while reducing liver enzymes and lipid peroxidation. The expression of apoptosis-related genes, such as Bax and caspase-3, was upregulated, whereas Bcl2 was significantly downregulated (*p* < 0.05). Furthermore, the expression of GPx and SOD genes in the liver was notably increased compared to the control group. The findings suggest that the phytosomal encapsulation of the phenolic rich fraction derived from *Allium ampeloprasum* L. can enhance the bioavailability of natural phytochemicals and improve their antitumor properties. The development of PRF-NPs as a nano drug delivery system holds promise for effective breast cancer treatment.

## 1. Introduction

Breast cancer is by far the most frequently diagnosed cancer worldwide which represents ¼ of all cancer cases as diagnosed in females. New cases of breast cancer were reported at 2.3 million in 2020 with approximately 685,000 deaths, and is projected to reach beyond 3 million new incidences and 1 million death cases by 2040 [1]. In the body, certain molecules called reactive oxygen species (ROS) and reactive nitrogen species (RNS) are normally produced as part of the defense system and as the by-products of cellular metabolic-processe-utilizing oxygen [2]. These reactive species include free radicals or certain molecules which may be oxidizing agents or convertible to free radicals. A wide variety of free radicals and other reactive oxygen species such as super oxide anion, hydroxyl radicals and nitric oxide can be formed in the human body and in food systems. ROS/RNS result in cell death by nonphysiological (necrotic) or regulated pathways (apoptotic) [3]. Despite the substantial advances made so far in combating breast cancer, techniques such as conventional chemotherapy have drawbacks and side effects and may develop chemoresistance in patients [4]. Natural bioactive compounds have been demonstrated to be efficient, safe, and cost-effective chemotherapeutic agents. Over 60% of anticancer agents that are currently in use are primarily derived from natural sources, particularly of plant origins [4]. Bioactive compounds such as phenolics can act as preventive and curative substances in cancer therapy without side effects [5]. The challenge to utilize phenolic compounds are, however, stability and low absorption rates [6,7]. Encapsulation methods in drug delivery systems have been shown to increase bioavailability and improve the anticancer efficacy of bioactive compounds including phenolics [7,8].

Phytosome-based phyto-phospholipid complexes represent a novel strategy aimed at enhancing the bioavailability of active substances. These complexes have emerged as a promising nano vesicular drug delivery system specifically designed for targeted tumor therapy [9]. Phytosomes are biocompatible mixtures of bioactive molecules and phospholipids that self-assemble into vesicular structures in aqueous environments; consequently, it enables the transportation of bioactive compounds derived from plants through the cell membrane. Electrostatic interactions and hydrogen bonding allow hydrophilic medicines to bind to the phospholipid heads, i.e., the polar ammonium and negative phosphate groups [10]. The high carrier loading capacity, stability, simple storage option in a solid ready-to-reconstruct state, small size, and biocompatibility makes phytosomes ideal carriers for highly water soluble medicines [11]. 

Wild leek (*Allium ampeloprasum* L.) is frequently found and grown in most parts of the world. Leeks are well known for their nutritional and therapeutic properties and offer a rich source of bioactive phytochemicals with multiple physiological functions, i.e., antiasthmatic, antidiabetic, antiplatelet, and antiatherosclerotic characteristics [12,13]. The present research aimed to synthesize and characterize the nano-phytosome-loaded phenolic rich fraction (PRF) of *Allium ampeloprasum* L. and to investigate the antitumor properties in mice bearing TUBO cells.

## 2. Materials and Methods

### 2.1. Plant Material

*Allium ampeloprasum* L. were collected on 15 May 2020 from Kalat highland, Karaj, Iran, identified in the Institute of Medicinal Plant, ACER, Karaj, and deposited (voucher specimen 003-IMPHM) in the Institute of Medicinal Plant Herbarium. Solvents, reagents, and chemicals were purchased from Dae Jung Co., Ltd. (Busan, Republic of Korea).

### 2.2. Extraction

The fresh leaves of *Allium ampeloprasum* L. were separated, washed, and dried. A 0.5 g dried sample was transferred to a 100 mL conical flask and mixed with 40 mL methanol 80% (*v/v*), followed by 10 mL of 6 M HCl, as previously described [14]. The mixture was subjected to agitation, after which it was transferred to a flask where it was heated to 90 °C for 2 h. Subsequently, the mixture was filtered using Whatman No.1 filter paper (Whatman, England). The filtrate was dried at 60 °C using a vacuumed Rotary Evaporator (Buchi, Switzerland). 

### 2.3. Phenolic Rich Fraction (PRF) Preparation

The extract was partitioned into different fractions successively with hexane, trichloromethane, ethyl ethanoate, butan-1-ol, and H_2_O [15]. Each extraction was carried out in triplicate using 200 mL fresh solvent. The filtered extract was concentrated using a vacuum rotary evaporator and Whatman No.1 filter paper (Buchi, Switzerland). Finally, the total phenolic compound (TPC) content per fraction was determined by adding 0.2 mL of the extract, 5 mL Folin–Ciocalteu reagent (1:10 *v/v*), and 4 mL 7.5% sodium carbonate to an aluminum-foil-covered test tube. Vortexed test tubes were used to determine the absorbance at 765 nm [16]. The fraction with the highest concentration of phenolic compounds was subsequently referred to as the phenolic rich fraction (PRF).

### 2.4. Nano-Phytosome Preparation and Characterization

In general, nano-phytosomes are made by reacting phospholipids with an active ingredient in an aprotic solvent [17]. Nano-phytosomes of PRF were synthesized in a 3:1 ratio in ethyl acetate. The compound was shaken overnight at 55–60 °C to completely dissolve. Following the removal of the solvent, nano-phytosomes were formed. The average size and stability of the particles, as well as their zeta potential, were assessed using the dynamic light scattering (DLS) technique. The morphology of the nano-phytosomes was determined using field emission scanning electron microscopy (FESEM).

### 2.5. Phenolic Profiling of Nano-phytosome

The quantification of phenolic compounds within the nano-phytosome was carried out using HPLC, following the previously established method [18]. Gallic acid, syringic acid, vanillic acid, salicylic acid, caffeic acid, pyrogallol, catechin, cinnamic acid, ellagic acid, naringin, chrysin, and ferulic acid were used as standards. HPLC column was eluted for 20 min prior to sample injection and equilibrated with 85% deionized water (solvent A) and 15% acetonitrile (solvent B). A gradient elution was used as follows: 85% (A) and 15% (B) for 60 min and 15% (A) and 85% (B) for 5 min, after which solvent B dropped to 15% (at 65 min) and maintained for another 5 min. A total of 70 min with flow rate 1 mL/min and absorbance at 280 nm were detected using an analytical column (Intersil ODS-3 5 um 4.6150 mm Gl Science Inc., Tokyo, Japan).

### 2.6. Animal Trials

For the antitumor properties of PRF and PRF loaded with phytosomes, 21 female BALB/c mice (average age 28 days, average body weight 19 g) were employed. Following a one-week adaptation period, the mice were randomly divided into three groups and housed in pairs in cages (30 × 15 × 15 cm) in a standard climatic room with controlled conditions (22 ± 2 °C, 50% humidity, and a 12:12 light/dark cycle). Animals were provided with unrestricted access to food and tap water. The TUBO breast cancer cell line was cultured in DMEM medium supplemented with 8% penicillin/streptomycin. A total of 5 × 10^5^ TUBO breast cancer cells suspended in 50 μL of PBS buffer were subcutaneously injected into all mice. Once the tumor reached approximately 3 mm in diameter, the treatments were administered daily for 28 days as follows: a negative control group (T1), mice treated with PRF (150 mg TPC/Kg/BW) (T2), and mice treated with nano-phytosome-loaded PRF (50 mg TPC/Kg/BW) (T3). The treatments were administered orally through gavage. Throughout the study, all groups received conventional and balanced meal pellets. Sunflower oil was used as a vehicle for administering PRF and PRF-NPs. The negative control group received corn oil without PRF and phytosomes. Weekly measurements of tumor sizes were taken and the tumor volume was calculated using the equation: [(long×width×height) × 0.52]. The amount of food consumed and changes in weight were monitored on a weekly basis. All methods and procedures were carried out in accordance with the relevant guidelines and regulations. All methods are reported in accordance with the ARRIVE guidelines.

### 2.7. Sample Collection

Animals were euthanized using pentobarbital-HCl (50 mg/kg, i.p.) and then dissected. Blood samples were collected from the abdominal aorta to measure the levels of alanine aminotransferase (ALT), alkaline phosphatase (ALP), and aspartate transaminase (AST) in the serum. Liver and tumor tissues were isolated for histological and gene expression analyses. For histological evaluation, tissues were immersed in 9.5% formalin for 56 h. Afterwards, the tissue fragments were dehydrated in a series of ethanol solutions, fixed in paraffin, and segmented into slices with a thickness of 4–5 mm using an automated microtome. These sections were stained with hematoxylin and eosin (H&E) and examined under a light microscope. For gene expression tests, tissue segments (50–100 mg) were extracted and immediately frozen in liquid nitrogen before being stored at −80 °C. The level of malondialdehyde (MDA) in the liver tissue was used as an indicator of oxidative stress. To measure MDA, the liver tissue was homogenized, and 200 µL of the resulting lysate was mixed with deionized water, BHT, sodium dodecyl sulphate, and thiobarbituric acid. The solution was then heated at 90 °C for 60 min and combined with *n*-butanol before being centrifuged. The absorbance of the resulting solution was recorded at 523 nm, and the percentage change in malondialdehyde (MDA) was reported.

### 2.8. Gene Expression Analysis

An extraction kit was utilized for the extraction of total RNA. The quantity and quality of the isolated RNAs were assessed using a spectrophotometer (Bio-Tek Instruments, Bad Friedrichshall, Germany). To synthesize cDNA, the cDNA synthesis kit was employed, using 5 g of RNA at 42 °C for 1 h. Amplification of the synthesized cDNA was carried out in 36 cycles using a thermocycler. Each reaction mixture consisted of 10 µL 2× Master Mix, 10 pmol primers, 1 µL template cDNA, and 8.5 µL dH_2_O, totaling 20 µL. The primer sequences for the genes of interest can be found in Table 1. The levels of mRNA expression for the selected genes were normalized to GAPDH expression level (used as a housekeeping gene for human genes). The relative expression of the investigated genes was determined using the 2^−Ct^ technique.

### 2.9. Statistical Analysis

The data were presented as mean ± standard deviation (SD) or standard error of the mean (SEM). To compare the means among different groups, the statistical analysis was conducted using the one-way ANOVA: post hoc Tukey test. The data analysis was performed using SPSS (version 19, IBM, New York, NY, USA). The α was set at 0.05 to identify statistically significant results.

## 3. Results and Discussion

### 3.1. Fractionation Analysis

The evaluation of the different solvent polarity fractions of *Allium ampeloprasum* L. revealed variations in the total phenolic compound (TPC) content. Among the fractions tested, the ethyl acetate fraction exhibited the highest TPC value of 263.5 ± 1.78 mg GAE/g DW of extract. In comparison, the TPC values for the *n*-butanol, water, chloroform, and hexane fractions were 179.1 ± 2.95, 81.2 ± 4.8, 37.6 ± 5.47, and 19.7 ± 5.28 mg GAE/g DW of extract, respectively. Based on the higher phenolic content, the ethyl acetate fraction was chosen for further experiments, including phytosome synthesis.

The observed variations in TPC among the different solvent fractions can be attributed to the selective extraction of bioactive compounds with varying polarities. Previous studies have demonstrated that the structure of bioactive compounds influences their solubility and separation [19]. Phenolic compounds, such as flavonoids and phenolic acids, possess different degrees of conjugation and the presence of hydroxyl groups, which contribute to their solubility in specific solvents [19,20].

The selection of an appropriate solvent polarity is crucial for the efficient extraction and separation of a large quantity of phenolic compounds. Solvent polarity determines the interactions between the solvent and the target compounds, affecting their solubility and partitioning. By choosing ethyl acetate as the solvent, which has intermediate polarity, a higher extraction efficiency for phenolic compounds with moderate polarity may be achieved [21]. Ethyl acetate has been widely utilized for the extraction of phenolic compounds from various plant sources due to its ability to dissolve a broad range of bioactive compounds [21].

The selection of the ethyl acetate fraction for phytosome synthesis is well founded, as it possesses the highest phenolic content. Phytosomes are phospholipid complexes formed between bioactive compounds and phospholipids, which enhance their bioavailability, stability, and therapeutic efficacy [22]. The higher phenolic content in the ethyl acetate fraction suggests a greater potential for encapsulation within the phytosome structure, providing an efficient delivery system for these compounds.

### 3.2. Phytosome Synthesis and Characterization

The size, morphology, and zeta potential of the nano-phytosome structure were evaluated. The PRF-NPs derived from *Allium ampeloprasum* L. exhibited an average particle size of 226 nm, along with a polydispersity index of 0.33, indicating uniformity. The zeta potential measurement recorded a value of −48.65, indicating a stable structure. Furthermore, the photomicrograph of the PRF-NPs confirmed their spherical shape, confirming the desired surface morphology (Figure 1). The high-performance liquid chromatography (HPLC) analysis of the PRF-loaded nano-phytosome identified naringin (975.6 µg/g) as the primary phenolic compound present in the nano-phytosome formulation (Table 2).

The size, morphology, and zeta potential of nano-phytosome structures play crucial roles in their stability, cellular uptake, and therapeutic efficacy. In this study, the PRF-NPs derived from *Allium ampeloprasum* L. were thoroughly characterized to understand their physicochemical properties.

The average particle size of 226 nm indicates that the PRF-NPs are in the nano-scale range. Nano-particles of this size have been reported to possess enhanced stability, prolonged circulation time, and improved cellular internalization [23,24]. The uniformity of the particle size distribution, as indicated by the low polydispersity index of 0.33, further supports the homogeneity of the formulation. These findings are consistent with previous studies that have highlighted the importance of particle size in determining the pharmacokinetics and biodistribution of nano-particles [24,25].

The zeta potential measurement of −48.65 demonstrates a high negative charge on the surface of the PRF-NPs. This negative charge is attributed to the presence of ionized functional groups on the surface, which contributes to the electrostatic stabilization of the nano-particles. The stable zeta potential value suggests that the nano-particles are less prone to aggregation or flocculation. This is crucial for maintaining the integrity and stability of the nano-phytosome structure during storage and administration [25].

The photomicrograph of the PRF-NPs confirms their spherical shape, which is a desirable characteristic for efficient cellular uptake and drug delivery. Spherical nano-particles have been shown to have improved circulation time, increased surface area for drug loading, and enhanced interaction with target cells [26]. The observed spherical morphology in this study suggests that the formulation process successfully yielded nano-particles with the desired shape.

Furthermore, the HPLC analysis of the PRF-loaded nano-phytosome revealed the presence of naringin as the primary phenolic compound. Naringin is a well-known bioactive flavonoid with various therapeutic properties, including antioxidant, anti-inflammatory, and anticancer effects [27]. Its identification in the nano-phytosome formulation suggests that the encapsulation process effectively retained the bioactive compounds from *Allium ampeloprasum* L. within the nano-particles.

### 3.3. Body Weight and Food Intake Analyses in Animals

The findings presented in Table 3 provide insights into the effects of PRF and PRF-NPs derived from *Allium ampeloprasum* L. on body weight and food intake in mice. The control group (T1) fed a normal diet exhibited the lowest weight gain and food intake compared to the other groups. However, the administration of PRF-NPs at a concentration of 50 mg TPC/Kg/BW significantly improved both the daily weight gain and feed intake in TUBO tumor-bearing mice.

Weight loss is a critical factor in cancer progression and has been associated with increased cancer risk and mortality [28]. Previous studies have demonstrated the impact of various interventions on weight parameters in cancer models. For instance, Gad et al. [29] conducted a study on Balb/c mice with induced human colorectal cancer and reported a significant decrease in body weight gain, food intake, and food efficiency ratio compared to the control group. Conversely, the administration of extracts from Pinus roxburghii and Nauplius graveolens led to notable improvements in these parameters.

The mechanism underlying the beneficial effects of PRF and PRF-NPs on body weight in tumor-bearing mice is complex and multifactorial. It could be attributed to the bioactive compounds present in *Allium ampeloprasum* L., which may exert regulatory effects on appetite, metabolism, and energy balance. Phytochemicals such as phenolic compounds have been reported to modulate various signaling pathways involved in food intake regulation, energy expenditure, and lipid metabolism [30]. These compounds may interact with the receptors and enzymes involved in appetite control and nutrient metabolism, thereby influencing weight gain and food intake.

Moreover, the antioxidant and anti-inflammatory properties of the bioactive compounds in *Allium ampeloprasum* L. could also contribute to the observed effects on body weight and food intake. Oxidative stress and inflammation have been implicated in cancer cachexia, a multifactorial syndrome characterized by weight loss and decreased appetite. The antioxidant and anti-inflammatory activities of phenolic compounds may help mitigate these pathological processes, thereby preserving body weight and promoting a healthy appetite.

### 3.4. Liver Enzyme and Lipid Peroxidation Assessment

The levels of liver enzymes, namely alanine aminotransferase (ALT), aspartate aminotransferase (AST), and alkaline phosphatase (ALP), were used as biochemical markers to assess hepatotoxicity. Hepatocellular damage is characterized by a substantial increase in ALT, AST, and ALP levels, as well as lipid peroxidation. The administration of PRF and PRF-NPs from *Allium ampeloprasum* L. at a dose of 50 TPC/Kg/BW in mice resulted in a significant reduction in ALT, ALS, ALP, and lipid peroxidation (Table 4). Notably, the impact of PRF-NPs was more pronounced (*p* < 0.05) in mitigating these parameters compared to PRF alone. These findings align with an earlier report [29], which indicated that liver enzyme levels increased in the positive control group, indicative of liver tissue damage. Conversely, it was revealed that extracts of *P. roxburghii* and *N. graveolens* demonstrated a significant reduction in liver enzymes, lipid peroxidation, and urea levels [29,31].

### 3.5. Histopathology, Size, and Weight of Tumor-Bearing Mice

The histological observations from each group (Figure 2) revealed distinct characteristics. In the positive control group (T1), tumor tissues appeared solid, dense, and compact. In contrast, the group treated with PRF and PRF-NPs from *Allium ampeloprasum* L. exhibited improved density of tumor tissues, and necrosis was more widespread, confirming the potent impact on cancer cells. These findings were consistent with the tumor weight and size measurements (Table 5). The T1 group demonstrated significantly (*p* < 0.05) higher tumor weight and larger size compared to the T2 and T3 groups. Notably, the effect of PRF-NPs was more pronounced, indicating strong antiproliferative effects. It was established that there is a strong association between an increased risk of cancer and the presence of a large tumor volume as well as the survival rate [32].

Of particular interest was the more pronounced effect observed with PRF-NPs, suggesting stronger antiproliferative effects compared to PRF alone. The enhanced efficacy of PRF-NPs could be attributed to the improved delivery and bioavailability of the active compounds to the tumor site, resulting in a more targeted and potent anticancer effect.

### 3.6. Antioxidant Gene Expression Pattern Analysis

The expression levels of antioxidant genes, specifically SOD and GPx, were evaluated as biomarkers of antioxidant activity [33]. Table 6 demonstrates the effectiveness of the line of defense that is continuously generated in normal body metabolism. For instance, the significantly improved expression of antioxidant genes was observed in the group treated with PRF and PRF-NPs from *Allium ampeloprasum* L. compared to the intact group (T1) (*p* < 0.05). This indicates that the treatment with PRF and PRF-NPs led to enhanced antioxidant gene expression in the liver.

The body is constantly exposed to free radicals that can cause damage. However, this damage can be mitigated through the action of antioxidant enzymes such as GPx, SOD, and CAT. These enzymes play a critical role in scavenging free radicals and protecting the body against oxidative stress. They form an integrated system of antioxidant defense that is closely associated with reactive oxygen species (ROS) [34].

### 3.7. Anticancer Gene Analysis in Mice Tumors

The expression levels of key cancer-related genes, including Bax, caspase-3, and Bcl2, were evaluated in mice tumors that received different treatments. The results are summarized in Table 7. The findings revealed that both PRF and PRF-NPs from *Allium ampeloprasum* L. upregulated the expression of Bax and caspase-3, while downregulating the expression of the Bcl2 gene in a dose-dependent manner compared to the control group. Notably, the impact of PRF-NPs was more significant and superior compared to PRF. Previous experiments have demonstrated the crucial role of natural bioactive compounds, such as phenolic constituents, in modulating the expression of anticancer genes, both upregulating and downregulating their activity.

The phytochemical analysis of nano-phytosomes loaded with phenolics from *Allium ampeloprasum* indicated a rich composition of gallic acid, pyrogallol, ferulic acid, ellagic acid, syringic acid, and naringin. Similar studies, for example, Moradzadeh et al. [28], have shown that epicatechin increases the Bax/Bcl-2 ratio, upregulates p53, p21, caspase-3, and caspase-9, and downregulates Bcl-2 in a human breast cancer cell line (T47D). Another compound, gallic acid, has been found to notably suppress cancer cell growth through the upregulation of Bax and downregulation of Bcl-2 [35,36].

These findings suggest that the administration of PRF and PRF-NPs from *Allium ampeloprasum* can influence the expression of key cancer-related genes, promoting apoptosis and inhibiting the development of cancer cells. The presence of specific phenolic compounds in the nano-phytosomes may contribute to such effects, as observed in previous studies as well [37,38,39].

## 4. Conclusions

This study synthesized a new type of nano-phytosomes containing various bioactive phytochemicals, including gallic acid, pyrogallol, ferulic acid, ellagic acid, syringic acid, and naringin. These nano-phytosomes demonstrated significant improvements in cellular redox state and induced apoptosis-mediated proliferation by regulating the expression of Bax, caspase-3, and Bcl2 genes. Overall, the administration of PRF and PRF-NPs derived from *Allium ampeloprasum* L. resulted in enhanced body weight gain and increased food intake in mice with tumors. These effects can be attributed to the presence of bioactive compounds that regulate appetite, metabolism, and inflammation. Further investigations are needed to understand the specific mechanisms and molecular targets involved in these effects. Nevertheless, these findings provide encouraging evidence for the potential utilization of *Allium ampeloprasum* L. formulations in managing weight loss and improving the nutritional status of cancer patients. The findings in the present study suggest that PRF-NPs derived from *Allium ampeloprasum* L. hold promise as a natural therapeutic agent for breast cancer treatment, effectively suppressing tumor growth.

## Figures and Tables

**Figure 1 genes-14-01547-f001:**
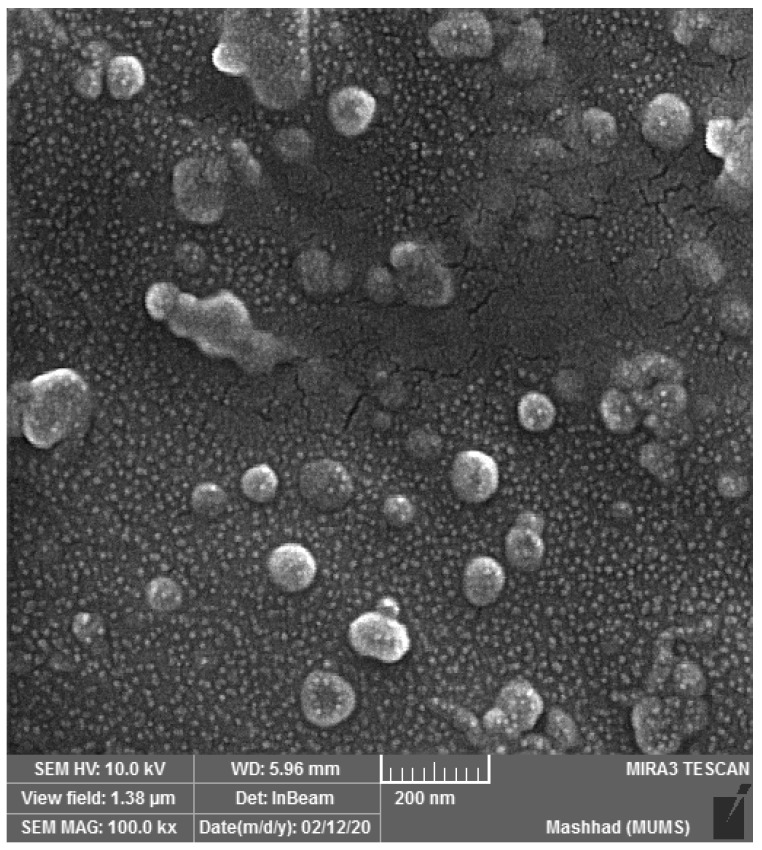
The FESEM analysis of PRF-NPs of *Allium ampeloprasum* L.

**Figure 2 genes-14-01547-f002:**
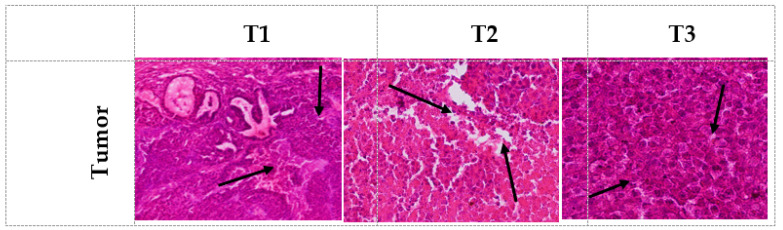
The histopathological alteration in the TUBO tumor-bearing mice during experiments for different treatments. T1: control, T2: mice receiving PRF (50 mg TPC/Kg/BW), T3: mice receiving nano-phytosome-loaded PRF (50 mg TPC/Kg/BW).

**Table 1 genes-14-01547-t001:** Primer sequences used in this study.

Genes		Forward (5′ to 3′)	Reverse (5′to 3′)
**Liver**	SOD	GAGACCTGGGCAATGTGACT	GTTTACTGCGCAATCCCAAT
	GPX	GTCCACCGTGTATGCCTTCTCC	TCTCCTGATGTCCGAACTGATTGC
	GAPDH	TGTGTCCGTCGTGGATCTGA	TTGCTGTTGAAGTCGCAGGAG
**Tunor**	bax	TTTGCTTCAGGGTTTCATCCA	CTCCATGTTACTGTCCAGTTCGT
	bcl2	CATGTGTGTGGAGAGCGTCAAC	CAGATAGGCACCCAGGGTGAT
	caspase-3	CTGGACTGTGGCATTGAGAC	ACAAAGCGACTGGATGAACC
	GAPDH	CCGGATCGACCACTACCTGGGCAAC	GTTCCCCACGTACTGGCCCAGGACCA

**Table 2 genes-14-01547-t002:** Phenolic compounds present in the PRF-NPs of *Allium ampeloprasum* L.

Phenolic Compounds (µg/g)		
**Gallic acid**	**Pyrogallol**	**Ferulic acid**
129.5 ± 1.7	203.8 ± 3.1	431.5 ± 4.1
**Ellagic acid**	**Syringic acid**	**Naringin**
811 ± 6.7	507.2 ± 3.5	975.6 ± 5.2

**Table 3 genes-14-01547-t003:** The averages of mice body weight changes and feed intake during the experiment for different treatments.

Average	T1	T2	T3	SEM
Average daily weight gain (mg)	163.5 ^c^	185.2 ^b^	228.6 ^a^	8.21
Average daily feed intake (g)	2.95 ^c^	3.21 ^b^	3.35 ^a^	0.04

T1: control, T2: mice receiving PRF (50 mg TPC/Kg/BW), T3: mice receiving nano-phytosome-loaded PRF (50 mg TPC/Kg/BW). Different letters in the same row indicate a significant difference (*p* < 0.05). The analyses were performed in triplicates.

**Table 4 genes-14-01547-t004:** The results of liver enzyme analysis and lipid peroxidation in the liver tissue.

Parameters	T1	T2	T3	SEM
ALT (IU/L)	183 ^a^	177 ^b^	155 ^c^	9.4
AST (IU/L)	76 ^a^	63 ^b^	54 ^c^	4.2
ALP (IU/L)	489 ^a^	407 ^b^	382 ^b^	19.5
MDA * (%)	100.0 ^a^	83.46 ^b^	74.19 ^c^	3.2

T1: control, T2: mice receiving PRF (50 mg TPC/Kg/BW), T3: mice receiving nano-phytosome-loaded PRF (50 mg TPC/Kg/BW). Different letters in the same row indicate a significant difference (*p* < 0.05). * Expressed as malondialdehyde changes relative to the control group (T1). The analyses were performed in triplicates.

**Table 5 genes-14-01547-t005:** Tumor weight and size characteristics over 28 days for different treatments.

Tumor	T1	T2	T3	SEM
Weight (g)	3.1 ^a^	2.61 ^b^	2.37 ^c^	0.14
Size (mm)	26.03 ^a^	24.95 ^b^	22.53 ^c^	0.36

T1: control, T2: mice receiving PRF (50 mg TPC/Kg/BW), T3: mice receiving nano-phytosome-loaded PRF (50 mg TPC/Kg/BW). Different letters in the same row indicate a significant difference (*p* < 0.05). The analyses were performed in triplicates.

**Table 6 genes-14-01547-t006:** The changes in the expression of antioxidant-related genes in mice liver for different treatments.

Gene Expression (Fold Changes)				S.E.M
Genes	T1	T2	T3	
SOD	1.0 ^c^	1.6 ^b^	2.4 ^a^	0.08
GPx	1.0 ^c^	1.5 ^b^	1.8 ^a^	0.09

T1: control, T2: mice receiving PRF (50 mg TPC/Kg/BW), T3: mice receiving nano-phytosome-loaded PRF (50 mg TPC/Kg/BW). Different letters in the same row indicate a significant difference (*p* < 0.05). The analysis was performed in triplicates, S.E.M: standard error of the means.

**Table 7 genes-14-01547-t007:** The changes in the expression of apoptosis-related genes in mice tumors for different treatments.

	Gene Expression (Fold Changes)			S.E.M
Genes	T1	T2	T3	
**Upregulated genes**				
Bax	1.0 ^c^	2.69 ^b^	3.48 ^a^	0.32
Caspase-3	1.0 ^c^	1.76 ^b^	2.24 ^a^	0.18
**Downregulated gene**				
Bcl2	1.0 ^c^	2.27 ^b^	2.94 ^a^	0.11

T1: control, T2: mice receiving PRF (50 mg TPC/Kg/BW), T3: mice receiving nano-phytosome-loaded PRF (50 mg TPC/Kg/BW). Different letters in the same row indicate a significant difference (*p* < 0.05). The analyses were performed in triplicates, S.E.M: standard error of the means.

## Data Availability

Not applicable.

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
