# Peer review of "Enhancing Healthcare Outcomes and Modulating Apoptosis- and Antioxidant-Related Genes through the Nano-Phytosomal Delivery of Phenolics Extracted from Allium ampeloprasum"

_genes, 2023, doi:10.3390/genes14081547_

Round 1

Reviewer 1 Report

The author that examine nano particle including drug delivery system with phytosome (PRF-NPs) for cancer treatment. They found that nano-phytosomes-loaded phenolic rich fraction (PRF) from Allium ampeloprasum L in TUBO tumor-bearing mice, which can enhance antioxidant enzyme expression and enhance apoptosis pathway.

Some comments as following:

1. In the paper is not well written and real data presenting as western blot analysis.

2. In the Abstract have mention TUBO colon carcinoma, but in Material and Methods has mention UBO breast cancer cell line that they further checked the information.

3. In the reference, citation should provide the Journal name, but a lot is not.

4. In the Table 6 and -7 should be provided the western blotting analysis for checking the protein level because caspase-3 should checked the cleavage form.

5. In the final, in the Introduction should include some information on ROS oxidative stress response and cell death pathway.

In the paper is not well written and real data presenting as western blot analysis.

Reviewer 2 Report

This is an interesting study about using nano drug delivery system to apply phenolic compounds that were extracted from Allium ampeloprasum L to mice, and demonstrated its anti-tumor effect. Importantly, this anti-cancer extract has no major side effect on mice as indicated by their normal body weight. In summary, this study provides new insights into the development of anti-cancer agents from plants with less side effect. At the same time, there are a few key points that the authors should address in order to make this study suitable for publication.

Major Comments

1) Line 308: “The expression of SOD and GPx as the main biomarker of antioxidant genes in mice”.

Authors should provide citation to support the claim that SOD and GPx are the biomarker “of antioxidant genes”, with examples of their applications from literature.

2) Table 2 indicates that the PRF-NPs of Allium ampeloprasum L contains gallic acid and ellagic acid, which are the agents that display anticancer effects. It would be useful to discuss (and compare when possible) the anti-cancer efficacy of PRF-NPs, gallic acid, and ellagic acid in mice using the data generated in this study (for PRF-NPs) and the published data from literature (for gallic acid, and ellagic acid).

3) Table 7 shows up-regulation of Bax and caspase-3 in the T2 and T3 (higher than the control T1), suggesting the link of the PRF to the induction of apoptosis. However, this is the “indirect evidence”, as the tumor cells that display up-regulation of Bax and caspase-3 do not necessary undergo apoptosis. Therefore, the authors Authors can consider providing direct evidence for apoptosis induction, such as cleavage of caspase-3 (by antibody) or DNA damage (TUNEL assay), as these are the hallmarks of apoptosis.

Minor Comments

1) Table 3. “The averages of mice body weight changes and feed intake during experiment receiving different treatments”

Authors should show how long the mice was treated in the table legend.

2) Line 292 “necrosis was more widespread”.

Authors should mark (using arrow) on Figure 2 to show the locations of necrosis.

/

Reviewer 3 Report

In the present manuscript entitled "Healthcare improvement and regulation of apoptosis and antioxidant-related genes by nanophytosomes loaded phenolics from Allium ampeloprasum" the authors report the development of a nanophytosomes-loaded phenolic rich fraction (PRF) from Allium ampeloprasum L. and its evaluation in activity antitumor for breast cancer. The manuscript should be improved in detailing information and explanations about the experimental conditions, as described below. For all these reasons, the study was not considered acceptable for publication in the journal.

 - In Plant material, write the collection date of the plant.

- How were the leaves dried? That is, were they dried in a drying oven, in the sun or in the wind?

- Explain in the manuscript the reason for selecting only the leaves of the plant to extract the bioactive compounds.

- In “2.2. Extraction” the dried plant extract was obtained. In “2.3. Phenolic rich fraction (PRF) preparation” the authors report that the extract was fractionated using “a separating funnel with hexane, chloroform, ethyl acetate, n-butanol, and water respectively”. Please detail the description. In each organic solvent extraction, did you use water to form a two-phase system in the separatory funnel, or did you just use the solvents to solubilize the extract?

- In “2.3. Phenolic rich fraction (PRF) preparation” I think that instead of writing “The filtered extract was”, it should be “extract solutions were” or “extractive fractions were” because it refers to the solutions obtained from extractions with solvents.

- On line 91, improve your English on the phrase “was referred to as a”.

- In “2.6. Animal trials” please clarify some questions:

1) Why did you use only female mice in the experiments? Can the hormonal cycle of females compromise the results of the experiments?

2) Why did you use mice aged 28 days? They are considered too young for the experiment and there may be unreliable results because the organism of these animals is not fully developed. Usually 7-9 week old mice are used.

3) Are the authors sure that all animals had the same body weight (19 g)? There is usually a variation in weight between animals.

4) The manuscript does not contain authorization from the Ethics Committee for the use of animals in experiments. What was the regulatory norm adopted by the researchers to carry out these tests with animals?

Reviewer 4 Report

The manuscript presented for review is very interesting and certainly brings a lot of new information to research in the field of cancer prevention, but it should be corrected by the authors.

1. The title of the manuscript does not inform the reader that the content concerns the activity of the tested compounds in the field of potential treatment of breast cancer or cancers in general.

2. The aim of the work presented in the Introduction part of the mnauscript is described very briefly and poorly. It should be developed. The authors should specify the hypothesis and link it to the research problem. The authors should explain their rationale for starting such research and expand the introductory part a bit, as it is very short and general.

3. Changes in the expression of genes related to oxidative stress could be presented in the form of graphs, it would be clearer.

4. Some of the discussion is quite short and concise, although the authors could expand it a bit.

5. The conclusion part is very short. Conclusions must be clearly formulated and inspire further research. It should be stated why this tenatic was undertaken and provide suggestions for future experiments.

Round 2

Reviewer 3 Report

Dear author,

In the Plant Material section write the plant collection date "February 2020". This information is important so that other researchers can reproduce the experiment.

Author Response

Dear Reviewers

We would like to thank you for the valuable and helpful comments. After applying the comments, the quality of the manuscript improved significantly. We greatly appreciate reviewer's time spent in preparing comments on this manuscript. The corrections have been applied accordingly and listed below.

With respect

 Dear author,

In the Plant Material section write the plant collection date "February 2020". This information is important so that other researchers can reproduce the experiment.

The exact date of plant collection has been double checked and the information have been added to manuscript.
